# Mutational alterations in the QRDR regions associated with fluoroquinolone resistance in *Pseudomonas aeruginosa* of clinical origin from Savar, Dhaka

Md. Shamsul Arefin[1☉], Meftahul Jannat Mitu[1☉], Shomaia Yasmin Mitu[1], Azmery Nurjahan[2], Mir Mobin[1], Shamsun Nahar[1], Hasnain Anjum[1,3]*, M. Hasibur Rahman[1]*

1 Department of Microbiology, Jahangirnagar University, Savar, Dhaka, Bangladesh, 2 Department of Biotechnology and Genetic Engineering, Jahangirnagar University, Savar, Dhaka, Bangladesh, 3 Department of Microbiology, Primeasia University, Banani, Dhaka, Bangladesh

☉ These authors contributed equally to this study.
* anjumabir27@gmail.com (HA); hasiburku@juniv.edu (MHR)

## Abstract

Bacterial DNA gyrase and topoisomerase IV are the major targets of quinolone antibiotic, and mutational alterations in quinolone resistance determining regions (QRDR) serve as major mechanism of resistance in most bacterial species, including *P. aeruginosa.* The present investigation was aimed to study the molecular mechanism of fluoroquinolone resistance among clinical *P. aeruginosa* isolated from Dhaka, Bangladesh, including alterations in target sites of the antimicrobial action. Laboratory collection of 53 *P. aeruginosa* were subjected to conventional cultural and biochemical characterization, followed by molecular identification using 16S rDNA sequencing. Susceptibility to ciprofloxacin and levofloxacin was tested by disc diffusion method followed by MIC assay. Resistant isolates were analyzed for mutation in their QRDR regions of *gyrA* and *parC,* and subjected to PCR detection of plasmid mediated quinolone resistance (PMQR) genes *qnrA, qnrS* and *qnrB.* Among the isolates, 28% were found to be resistant to both fluoroquinolones tested. All of the fluoroquinolone resistant isolates carried a single mutation in *gyrA* (Thr-83-Ile), while 20% carried a single *parC* mutation (Ser-87-Leu). Higher level of MIC was observed in isolates carrying alterations at both sites. None of the isolates harbored any PMQR genes investigated, suggesting that chromosomal mutations in QRDR regions to be the major contributing factor for quinolone resistance in *P. aeruginosa* under investigation.

## Introduction

*Pseudomonas aeruginosa* is a cosmopolitan member of the Pseudomonaceae family commonly associated with opportunistic and nosocomial infections [1]. The recent emergence of multidrug-resistant and extensively-drug resistant strains of *P. aeruginosa* is alarming, indicating diminished options of therapeutically effective antibiotics [2]. The nosocomial pathogen can naturally resist a wide range of antibiotics through lower outer membrane permeability,

**Data availability statement:** All relevant data are within the manuscript and its Supporting Information files. Sequence data of gyrA and parC from a representative isolate is available at the GenBank databases under Accession numbers PP501828 (gyrA) and PP526740 (parC).

**Funding:** MHR received a grant from the Grants for Advanced Research in Education (GARE), the Ministry of Education, Bangladesh (Award ID: LS-2018661/Ref:bs-37.20.0000.004.033.020.2016.673) (Website: https://moedu.gov.bd). The funders did not play any role in the study design, data collection and analysis, decision to publish, or preparation of the manuscript.

**Competing interests:** The authors have declared that no competing interests exist.

multidrug efflux pumps and chromosomally encoded enzymes [2–4]. Antibiotics like carbapenems, fluoroquinolones, piperacillin, ceftazidime and aminoglycosides are commonly used for treatment of infection caused by *P. aeruginosa* [5]. However, emerging resistance to these antibiotics has been creating outbreaks of multidrug resistant isolates, which is becoming a major infection-related treatment burden [6]. Resistance to drugs like quinolone in *P. aeruginosa* increases difficulties in treating severe infections like sepsis and catheter-associated urinary tract infection (CA-UTI) [7].

Fluoroquinolone antibiotics, including ciprofloxacin and levofloxacin, can inhibit DNA Gyrase and Topoisomerase of *P. aeruginosa,* leading to bacteriostatic activity [8]. Resistance to these antibiotics is primarily mediated through mutational alterations in the Quinolone Resistance Determining Regions or QRDR motif [9]. In *P. aeruginosa,* fluoroquinolone target includes DNA Gyrase (GyrA and GyrB) and DNA Topoisomerase IV (ParC and ParE) molecular subunits, which are responsible for ATP-dependent cleaving and rebinding double-stranded DNA during replication [10]. Fluoroquinolones can bind to GyrA and/or ParC subunits and inhibit the catalytic effect of the protein, resulting inhibition of bacterial DNA replication [11]. Amino acid substitution in the 67-106[th] position of the GyrA subunit of *P. aeruginosa* has been reported as the major mechanism of resistance against the antibiotic, as it leads to a reduced binding affinity between fluoroquinolones and GyrA [12–13]. Evidence suggests that substitution of amino acid residues Thr83 of the QRDR motif in GyrA is essential for the development of quinolone-resistant *P. aeruginosa* [13–15]. Additional mutational alteration in the QRDR motif of ParC subunit of DNA topoisomerase IV has also been associated with increased level of quinolone resistance in *P. aeruginosa* [13].

Alongside mutational alteration in the QRDR motif of GyrA and ParC, other resistance mechanisms also significantly contribute to reduced quinolone sensitivity in *P. aeruginosa.* This includes overexpression of efflux molecular pumps like MexAB-OprM and MexCD-OprJ through mutational alterations in their regulatory genes *mexR* and *nfxB,* respectively [16], and acquisition of Plasmid Mediated Quinolone Resistance (PMQR) genes like *qnrA, qnrB, qnrC* and *qnrS* [17]. Surveillance of antimicrobial resistance in nosocomial pathogens like *P. aeruginosa* is essential to determine therapeutic interventions. Frequent dissemination of mobile genetic elements like PMQR genes in nosocomial infection from high complexity hospitals can be a major issue [17]. Since data on the resistance determinants in fluoroquinolone-resistant *P. aeruginosa* from Savar, Dhaka is scarce, the current investigation aimed to detect PMQR mediated fluoroquinolone resistance in *P. aeruginosa,* as well as the role of mutational alterations among the isolates. The objective of the present study was to investigate the DNA sequences of *P. aeruginosa* QRDR motifs to understand the association of fluoroquinolone tolerance level and alteration in GyrA and ParC, and the prevalence of PMQR variants in clinical *P. aeruginosa* isolates from Bangladesh.

## Materials and methods

### Bacterial isolates

Laboratory collection of 53 clinical *P. aeruginosa* collected from two renowned hospitals in Savar was investigated in this study. A total of 238 clinical samples including urine, pus, secondary wound infection swab, burn wound swab, catheter swab, blood and tracheal aspirate were collected from April 2020 to January 2021. Reconfirmation of the isolates' identity was conducted using conventional cultural and biochemical characterization following Bargey's Manual of Systemic Bacteriology [18–20]. Cultural identification was done on Cetrimide Agar (Scharlab SL, Spain), where colonies with green pigmentation and fluorescence under ultraviolet was characteristic of *P. aeruginosa.* As positive and negative controls for the

experiments, *P. aeruginosa* ATCC 27853 and *E. coli* ATCC 27853 were used, respectively. For further verification of their identity, fourteen representative isolates from different antibiotic resistance pattern were subjected to 16SrDNA sequencing using fd1 and rp2 primers (Table 1) and identity of the isolates was confirmed by BLAST analysis of the sequences using an online database (https://blast.ncbi.nlm.nih.gov/Blast.cgi).

## Screening of fluoroquinolone resistance in clinical *P. aeruginosa*

Fluoroquinolone susceptibility among the clinical *P. aeruginosa* isolates were determined by disc diffusion assay against Ciprofloxacin and Levofloxacin, followed by minimum inhibitory concentration (MIC) analysis by agar dilution method. Results were interpreted according to Clinical Laboratory Standard Institute (CLSI) standards 2021 (31st Edition) [21]. *P. aeruginosa* ATCC 27853 was used as control for this experiment.

## Sequence analysis of the QRDRs in GyrA and ParC

Fluoroquinolone resistant isolates were subjected to analysis of alterations in their GyrA and ParC subunits of DNA Gyrase and Topoisomerase IV, respectively. The amplification of putative QRDR region in *gyrA* and *parC* gene was carried out using primer sequence and conditions obtained from a previous study (Table 1) [22]. PCR-amplified products were purified using commercially available Wizard® SV Gel and PCR Clean-Up System from Promega, USA, following instructions provided. Purified PCR-amplified products were sequenced by the dideoxy chain-termination method [26]. Sequence alignment and amino acid alterations was analyzed by Bioedit Sequence Alignment Editor (version 7.0.5.3). The sequences were compared with associated *P. aeruginosa* PAO1 loci from NCBI GenBank as reference.

## PCR amplification of plasmid mediated quinolone resistance (PMQR) genes

The presence of three different variants of *qnr* genes namely *qnrA, qnrB* and *qnrS* were investigated in all isolates exhibiting phenotypic resistance against fluoroquinolone antibiotics. Locally curated strains *E. coli* DMC1 (for *qnrA* and *qnrB*) and *E. coli* Shishu3 (for *qnrS*) were used as positive control. The primers and annealing temperatures are enlisted in Table 1.

**Statistical analysis.** A validated questionnaire was used for data collection. Collected data were verified and analyzed by using IBM SPSS Statistics Data Editor (Version 21). STATA 15 was used for subsequent analysis. A *p-value* of < 0.05 was considered significant.

**Table 1. Primers used in this study.**

| Gene | Primer Sequence (5' to 3') | Use | Annealing Temperature (°C) | Amplicon Size/ Nucleotide Position | Reference |
|------|----------------------------|-----|----------------------------|------------------------------------|-----------|
| *gyrA* | GAC GGC CTG AAG CCG GTG CAC | Amplification and sequencing of *gyrA* subunit of Toposiomerase II | 65 | 115–135 | [22] |
|  | GCC CAC GGC GAT ACC GCT GGA |  |  | 531–511 |  |
| *parC* | CGA GCA GGC CTA TCT GAA CTA T | Amplification and sequencing of *parC* subunit of Toposiomerase IV | 55 | 63–84 |  |
|  | GAA GGA CTT GGG ATC GTC CGG A |  |  | 366–344 |  |
| *qnrA* | AGA GGA TTT CTC ACG CCA GG | Amplification of *qnrA* | 54 | 580 bp | [23] |
|  | TGC CAG GCA CAG ATC TTG AC |  |  |  |  |
| *qnrB* | CCT GAG CGG CAC TGA ATT TAT | Amplification of *qnrB* | 60 | 390 bp | [24] |
|  | GTT TGC TGC TCG CCA GTC GA |  |  |  |  |
| *qnrS* | GCA AGT TCA TTG AAC AGG GT | Amplification of *qnrS* | 54 | 428 bp | [23] |
|  | TCT AAA CCG TCG AGT TCG GCG |  |  |  |  |
| fd1 | TCT AAA CCG TCG AGT TCG GCG | Amplification and sequencing of 16S rDNA | 42 | 1500 bp | [25] |
| rp2 | ACG GCT ACC TTG TTA CGA CTT |  |  |  |  |

**Ethics statement.** This study was approved by the Ethics and Research Review Committee of the Jahangirnagar University Faculty of Biological Sciences [Ref No: BBEC, JU/M 2020 (1)4]. Written informed consent was obtained from patients for sample collection, and their personal identities along with other information were anonymized.

# Results

## Fluoroquinolone resistance in *P. aeruginosa*

All 53 isolates of *P. aeruginosa* were subjected to fluoroquinolone susceptibility testing against ciprofloxacin and levofloxacin by disk-diffusion method followed by MIC assay. Among them, 15 (28%) were found to be resistant to both ciprofloxacin and levofloxacin, and all resistant isolates exhibited a high MIC to the fluoroquinolones tested ($\geq$16 to $\geq$128 µg/ml).

## Sequence analysis of QRDR regions

The amino acid alterations in GyrA and ParC QRDR regions of 15 fluoroquinolone resistant *P. aeruginosa* were analyzed by sequencing and comparing with the corresponding sequences of *P. aeruginosa* PAO1. According to the pattern of amino acid alteration, the isolates were categorized into two distinct groups. Group I consisted of isolates with a single mutation at Thr-83-Ile in *gyrA*, while Group II contained one mutation at Thr-83-Ile in *gyrA* and one mutation at Ser-87-Leu in *parC* (Table 2). Among the 15 isolates, all carried a single mutation (Thr-83-Ile) in *gyrA* (Fig 1). Single mutation in *parC (*Ser-87-Leu) was also found in 3 of 15 isolates (19%) (Fig 1). No additional mutations were observed in the QRDR regions. Sequence data of *gyrA* and *parC* from a representative isolate carrying mutations at both sites (PWS10) have been submitted to the GenBank databases under accession numbers PP501828 (*gyrA)* and PP526740 (*parC).*

Alteration in both *gyrA* and *parC* subunit revealed a statistically significant level of tolerance (MIC value $\geq$ 128 µg/ml) against both ciprofloxacin ($p = 0.002$) and levofloxacin ($p = 0.002$) (Table 3). Isolates with a single *gyrA* mutation had fluoroquinolone MIC ranges from 16 to 64 µg/ml.

## Occurrence of PMQR genes

PCR detection of three PMQR variants (*qnrA, qnrB* and *qnrS*) revealed that none of the 15 fluoroquinolone resistant isolates carried the PMQR genes (Fig 2).

# Discussion

Second and third generation fluoroquinolones like ciprofloxacin and levofloxacin holds immense importance in therapeutic intervention against *Pseudomonas aeruginosa,* as they are broad spectrum antibiotics that is often the drug of choice for oral treatment of patients in outpatient settings [27]. Additionally, fluoroquinolones are important members of effective combined therapeutic

**Table 2. Amino acid alterations in GyrA and ParC in fluoroquinolone resistant *P. aeruginosa* isolates.**

| Groups | No. of Isolates | Alterations in QRDR | |
|---|---|---|---|
| | | **GyrA at position** | **ParC at position** |
| | | **83** | **87** |
| PAO1 | | Thr (ACC) | Ser (TCG) |
| I | 12 | Ile (ATC) | – |
| II | 3 | Ile (ATC) | Leu (TTG) |

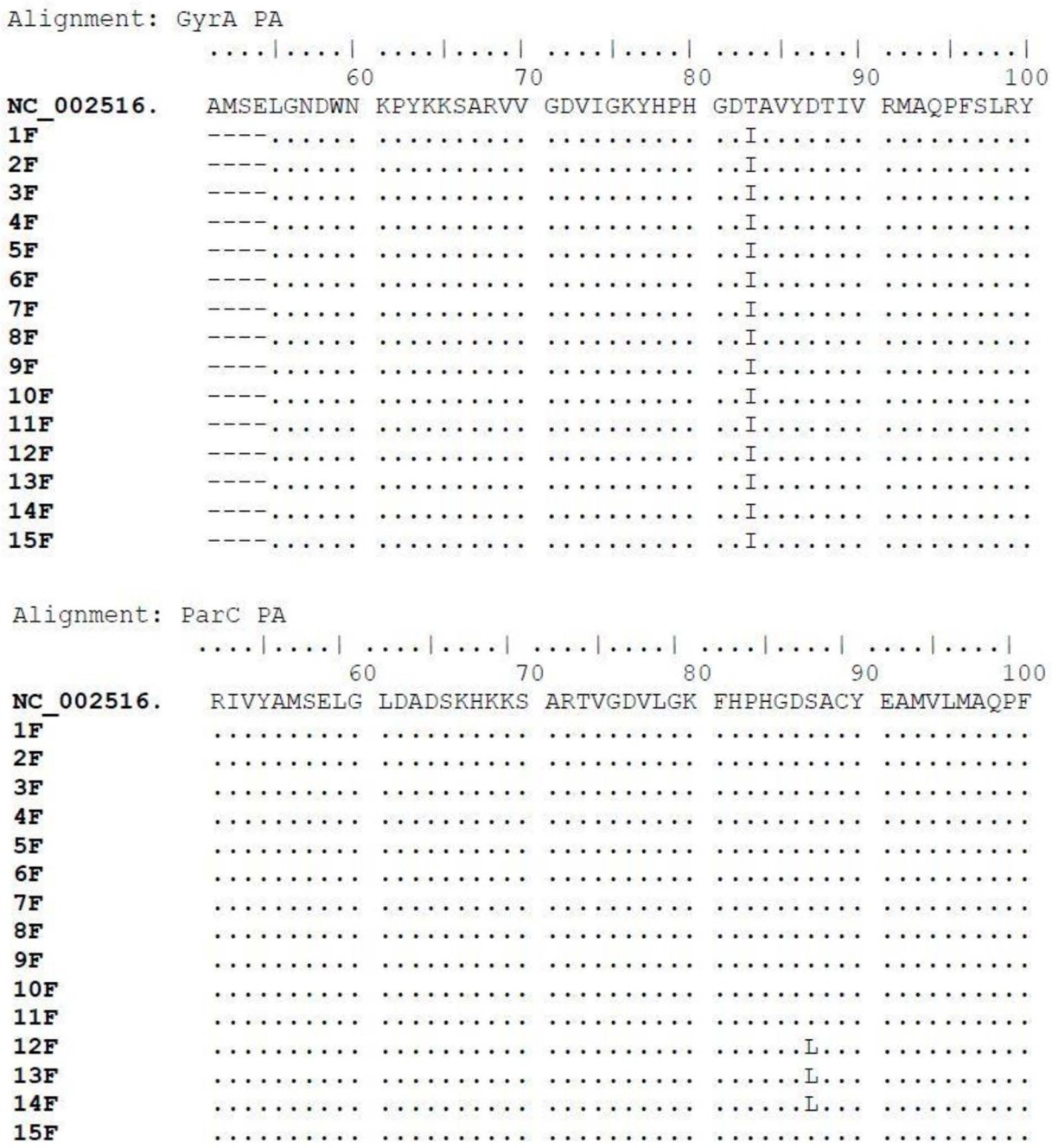

**Fig 1. Sequence analysis of *gyrA* and *parC* showing alterations; Alignment: GyrA PA shows Thr-83-Ile in *gyrA* of all isolates tested, and Alignment: ParC PA shows Ser-87-Leu in *parC* of 12F, 13F and 14F.**

strategy in critical cases like COPD (Chronic Obstructive Pulmonary Disease) exacerbations [28]. However, the rise of DTR (Difficult-to-Treat Resistant) *P. aeruginosa,* characterized by resistant to antibiotics like ceftazidime, cefepime, piperacillin-tazobactam, aztreonam, imipenem-cilastatin, meropenem, ciprofloxacin, and levofloxacin, raises concern as battling infections become increasingly difficult [29]. Understanding resistance mechanisms is a key part of this battle.

*P. aeruginosa* can employ multiple mechanisms to resist fluoroquinolones. Beside decreased level of antibiotic accumulation using lower outer membrane permeability and active molecular efflux pumps, alteration of QRDR motifs within topoisomerase II (GyrA and GyrB subunits) and topoisomerase IV (ParC and ParE subunits) has been considered to be the principal mechanism of fluoroquinolone resistance [30–34]. Several important pathogens, including *E. coli, Staphylococcus aureus, Streptococcus pneumoniae* and *Neisseria gonorrhoeae* has been found to have *gyrA* as their primary target of mutation alone or in combination with *parC* gene [14,35–37]. Additional mutation of *mexR, nfxC* or *nfxB* genes are also found to cause limited sensitivity to quinolones [13,38,39].

In *P. aeruginosa,* alteration at position 83 in *gyrA* is most commonly associated with high level of resistance [12,40]. Mutation from Threonine to a hydrophobic amino acid at this position tends to generally confer more resistance than other similar alterations at position 87, as observed in several reports [12–14]. The present study observed that all fluoroquinolone resistant isolates carried a single mutation at position 83, where threonine (Thr) was replaced with Isoleucine (Ile) (Table 2). The result is in accordance with previous studies, as it is reported as the most common site of mutation at the QRDR region [13,14,41,42]. Additional novel mutations in *gyrA* have also been reported, including Asp-87-Asn, Asp-87-Gly

**Table 3. Association of mutations in QRDR region and MIC value against fluoroquinolones.**

| Group | No. of Isolates | Fluoroquinolone antibiotics | No. of isolates with corresponding MIC (μg/ml) value | | | | | | | | | |
|---|---|---|---|---|---|---|---|---|---|---|---|---|
| | | | 0.125 | 0.5 | 1 | 2 | 4 | 8 | 16 | 32 | 64 | ≥128 |
| I; Single alteration in *gyrA* (83) | 12 | Ciprofloxacin | | | | | | | 5 | 5 | 2 | |
| | | Levofloxacin | | | | | | | 1 | 6 | 5 | |
| II; Alterations in *gyrA* (83) and *parC* (87) | 3 | Ciprofloxacin | | | | | | | | | | 3 |
| | | Levofloxacin | | | | | | | | | | 3 |
| III; Quinolone sensitive isolates | 38 | Ciprofloxacin | 12 | 25 | 1 | | | | | | | |
| | | Levofloxacin | | 26 | 10 | 1 | 1 | | | | | |

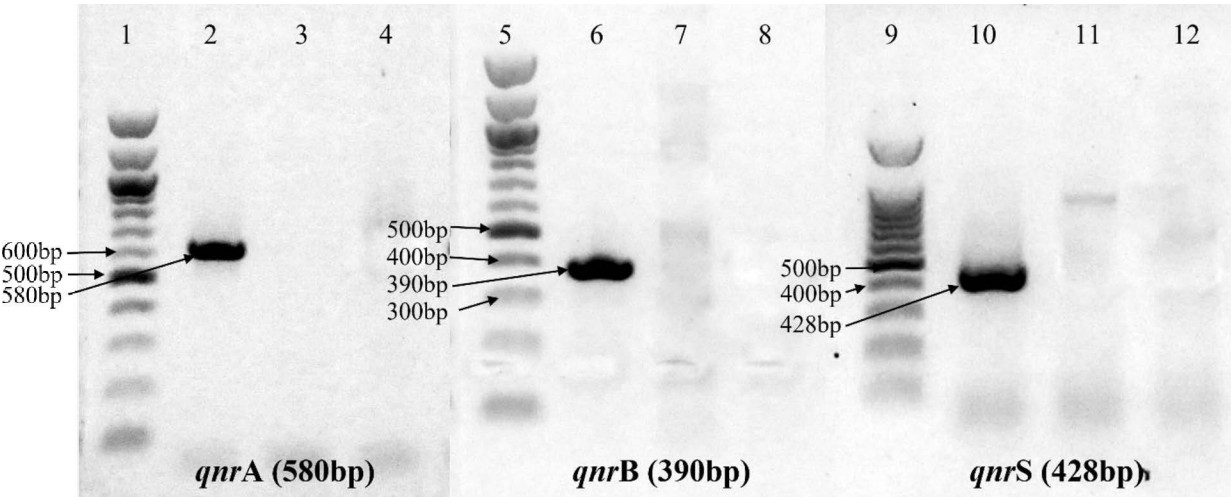

**Fig 2. PCR based detection of PMQR variants.** Variants *qnr* gene sequences were PCR amplified using specific primers (Table 1). PCR products were electrophoresed through 1% agarose gels, stained with ethidium bromide, and visualized with UV light. The 100 bp DNA ladder (Promega, USA; Catalogue number G2101) indicates the size of PCR products. Lane 2, 6 and 10 represent variant-specific positive controls, and lane 3, 7 and 11 represent negative controls. No isolates were positive in *qnr*A, *qnr*B and *qnr*S specific PCR.

and Gln-106-Leu, which often occur along with alterations at position 83 and associated with high MIC level [15,43,44]. However, none of the isolates in this study had such alterations at position 87 and 106. Although Thr-83-Ile mutation can result resistance to second-generation quinolones like ciprofloxacin and levofloxacin as observed in this study (Table 3), its effect has been seen to be limited on more recent generation of quinolones like sitafloxacin and clinafloxacin [39].

Substitution of Leu for Ser-87 in *parC* subunit was observed in three isolates. This substitution is closely associated with fluoroquinolone resistance, and has been reported as a second step mutation in isolates already having a single alteration of *gyrA* in *P. aeruginosa* [13,15]. This supports the present data, as the isolates with alteration Ser-87-Leu in *parC* also carried alteration Thr-83-Ile in *gyrA*. These isolates had an elevated level of both ciprofloxacin and levofloxacin MIC (≥128 μg/ml). Additional mechanism of resistance like overexpression of efflux pump MexAB-OprM by mutational alteration of *mexR* could also contribute to fluoroquinolone resistance in *P. aeruginosa* [39]. However, no phenotypic evidence of efflux pump overexpression was observed in efflux inhibitor induced MIC depression assay in this study, so analysis of mutation in *mexR* has not been conducted.

Although variants of PMQR genes like *qnrA, qnrB, qnrC, qnrD* and *qnrS* are not frequently found in *P. aeruginosa* [45], several recent studies have reported the increasing number of occurrences of PMQR genes in clinical *P. aeruginosa* [46–49]. But none of the quinolone resistant isolates from this study were seen to be harboring any of the three variants of *qnr* genes tested. Quinolone resistance through acquisition of PMQR genes is a mechanism commonly observed among the members of the *Enterobacteriaceae* family [50], and was first discovered by Martinez *et al.* that it could be transmitted to *P. aeruginosa* in vitro as well [51]. This is a major concern as acquisition of such resistance elements can further diminish the antimicrobial activity of fluoroquinolones [52].

There are some limitations to the present study. Additional QRDR regions of *P. aeruginosa* like *gyrB* and *parE* is not investigated for mutational alterations, and newer generations of fluoroquinolones like sitafloxacin are not studied as well. Future prospects of the investigation include an in-depth analysis of mutations in other significant regions and their association with the development of multidrug-resistant *P. aeruginosa*.

## Conclusion

Culmination of data from this study suggests mutational alteration at QRDR regions, especially GyrA is the most significant mechanism of resistance to fluoroquinolones. Although other mechanisms like acquisition of PMQR and overexpression of efflux pumps contribute to resistance, the results indicate that mutational alteration alone can lead to the development of resistance against second generation fluoroquinolones like ciprofloxacin and levofloxacin. Broad spectrum antibiotics like fluoroquinolones are one of the safest therapeutic options to treat infections caused by *P. aeruginosa.* However, the ever-increasing resistance to the antibiotics is a major concern. Due to the bacteria's incredible capacity in resisting various antibiotics, it is essential to raise awareness to implement antibiotic stewardship.

## Supporting information

**S1 File. Enclosing Sequence data of gyrA and parC.**
(ZIP)

**S2 File. Sample related data.**
(DOCX)

**S1_Raw_Images. Supplementary images.**
(DOCX)

## Acknowledgments

The authors would like to thank the laboratory personnel of Enam Medical College Hospital and Gonoshasthaya Medical College Hospital, Dhaka, Bangladesh, for their support in the collection of the clinical samples. The authors would also like to show gratitude for Dr. M. A. Karim Rumi for graciously revising and correcting the language issues of the manuscript.

## Author contributions

**Conceptualization:** M. Hasibur Rahman.

**Data curation:** Md. Shamsul Arefin, Meftahul Jannat Mitu, Azmery Nurjahan, Hasnain Anjum.

**Formal analysis:** Md. Shamsul Arefin, Meftahul Jannat Mitu, Shomaia Yasmin Mitu, Mir Mobin, Hasnain Anjum.

**Funding acquisition:** M. Hasibur Rahman.

**Investigation:** Md. Shamsul Arefin, Meftahul Jannat Mitu, Shomaia Yasmin Mitu, Azmery Nurjahan, Mir Mobin, Hasnain Anjum.

**Methodology:** Md. Shamsul Arefin, Meftahul Jannat Mitu, Shamsun Nahar, Hasnain Anjum, M. Hasibur Rahman.

**Resources:** M. Hasibur Rahman.

**Supervision:** Shamsun Nahar, M. Hasibur Rahman.

**Visualization:** Meftahul Jannat Mitu, Hasnain Anjum.

**Writing – original draft:** Hasnain Anjum.

**Writing – review & editing:** Shamsun Nahar, Hasnain Anjum, M. Hasibur Rahman.

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
