## [Decision Letter · Decision Letter 0]

30 Jul 2024

PONE-D-24-13068Mutational alterations in the QRDR regions associated with fluoroquinolone resistance in Pseudomonas aeruginosa of clinical origin from Savar, DhakaPLOS ONE

Dear Dr. Rahman,

Thank you for submitting your manuscript to PLOS ONE. After careful consideration, we feel that it has merit but does not fully meet PLOS ONE’s publication criteria as it currently stands. Therefore, we invite you to submit a revised version of the manuscript that addresses the points raised during the review process.

**Please go through the Reviewers comments carefully and address them accordingly.** Please submit your revised manuscript by Sep 13 2024 11:59PM. If you will need more time than this to complete your revisions, please reply to this message or contact the journal office at plosone@plos.org . Please include the following items when submitting your revised manuscript:

We look forward to receiving your revised manuscript.

Kind regards,

Furqan Kabir

Academic Editor

PLOS ONE

2. Please amend the manuscript submission data (via Edit Submission) to include author Azmeri Noorjahan.

3. Please amend your authorship list in your manuscript file to include authors Azmery Nurjahan, Shamsun Nahar and Shomia Yasmin Mitu.

Reviewers' comments:

Reviewer's Responses to Questions

**Comments to the Author**

1. Is the manuscript technically sound, and do the data support the conclusions?

Reviewer #1: Yes

Reviewer #2: Yes

2. Has the statistical analysis been performed appropriately and rigorously? 

Reviewer #1: Yes

Reviewer #2: No

3. Have the authors made all data underlying the findings in their manuscript fully available?

Reviewer #1: No

Reviewer #2: Yes

4. Is the manuscript presented in an intelligible fashion and written in standard English?

Reviewer #1: Yes

Reviewer #2: Yes

5. Review Comments to the Author

Reviewer #1: 1-the language needs revise by profisionals

2- please add at least 4 ref from middleeast

3- add statics to your study

4- whhre is the anaylsis of sequencing??

5- where is figurs of your study?

6- please add figure of your phnotypic test

Reviewer #2: Major comments:

- In table 3: You have to create another group without detected mutations mention the number of isolates at each MIC in table 3.

- Where is the statistical analysis that confirm the significant or non-significant correlation between the detected mutations and the resistance?

Minor comments:

- Please check the language carefully in the whole manuscript e.g. Line101: "were" and Line 36.

- Clarify the abbreviation "QRDR” in line 56, as it the first time to be mentioned in introduction and please clarify other abbreviations.

- In line 61, does the word " motif" refer to the QRDR motif ? please clarify.

- In methods, please mention the ethical approval details.

- Please carefully revise the whole manuscript as it contains some" review track changes" e,g line 78 ??

- Please mention the year of the CLSI version that was followed in this study in the references list.

- In figure 1: Indicate the molecular size of different bands of the marker and also indicate its catalogue number

- In table 1:

• Write the name of genes correctly "Italic and small" in column 3 and in the whole manuscript.

• Put the "�C" only in the heading.

- - Write the name of protein correctly "Not Italic and start with capital letter" in Table 2 and in the whole manuscript when it refers to protein but not gene.

6. PLOS authors have the option to publish the peer review history of their article (what does this mean? ). If published, this will include your full peer review and any attached files.

**Do you want your identity to be public for this peer review?** For information about this choice, including consent withdrawal, please see our Privacy Policy .

Reviewer #1: No

Reviewer #2: **Yes: ** Mohammed El-Mowafy

---

## [Author Response · Author response to Decision Letter 1]

13 Sep 2024

Author’s reply: Thank you so much. The file and author affiliation has been updated according to the style template.

2. Please amend the manuscript submission data (via Edit Submission) to include author Azmeri Noorjahan.

Author’s reply: Thank you so much. The author’s name has been included.

3. Please amend your authorship list in your manuscript file to include authors Azmery Nurjahan, Shamsun Nahar and Shomia Yasmin Mitu.

Author’s reply: Thank you so much. The authorship list has been updated accordingly.

Author’s reply: Thank you. All original uncropped and unadjusted images of gels used in the manuscript have been included in the Supporting information file. All requirements for gel reporting have been met. Images of phenotypic tests conducted and additional gel images are also included in the file as requested by Reviewer #1. I hope this is acceptable to you.

Author’s reply: Thank you so much. The issue has been corrected accordingly.

Author’s reply: Thank you so much. The ethics statement has been moved to the Methods section of the manuscript. I trust this is okay now.

Review Comments to the Author

Reviewer #1:

1-the language needs revise by profisionals

Author’s reply: Thank you for the comment. The language of the manuscript has been revised by an academician who is currently a faculty member at University of Kansas Medical Center and has over 100 publications in peer reviewed journals. We trust this is okay now.

2- please add at least 4 ref from middleeast

Author’s reply: Thank you for the suggestion. A study from the Middle East was already referenced in this manuscript, and three additional references of studies obtained from the Middle East have been added that discuss data similar/supportive of the present study. This includes references 30, 43, 44 and 45. I hope this is acceptable to you now.

3- add statics to your study

Author’s Reply: Thank you for the suggestion. Statistical analysis of significant parts of data has been added to the result section (Line 160 onwards). I hope this is acceptable to you now.

4- whhre is the anaylsis of sequencing??

Author’s Reply: Thank you. As per your comment, the analysis of sequence data has been reported through Fig 1 now, which depicts mutational alterations in both genes studied. I trust this is acceptable:

5- where is figurs of your study?

Author’s Reply: Thank you for your question. Figures relevant to this study have been uploaded separately, as per the instructions for figure submission by the journal. I hope this is in accordance to you.

6- please add figure of your phnotypic test

Author’s Reply: Thank you. The figures of phenotypic tests relevant to this study have now been uploaded as supporting information. I believe you can find the figures in Supporting Information File titled S1_raw_images, and trust this is acceptable to you.

Reviewer #2:

Major comments:

- In table 3: You have to create another group without detected mutations mention the number of isolates at each MIC in table 3.

Author’s reply: Thank you so much for your suggestion. Quinolone sensitive isolates with MIC ≤4 µg/ml were not subjected to sequence analysis, hence were not included in the table previously. We have now updated Table 3 as per your suggestion, and I hope this is acceptable to you. The updated table is attached below:

Table 3. Association of mutations in QRDR region and tolerance level against fluoroquinolones

Group No. of Isolates Fluoroquinolone antibiotics No. of isolates with corresponding MIC (µg/ml) value

0.125 0.5 1 2 4 8 16 32 64 128

I; Single alteration in gyrA(83) 12 Ciprofloxacin 5 5 2

Levofloxacin 1 6 5

II; Alterations in gyrA (83) and parC (87) 3 Ciprofloxacin 3

Levofloxacin 3

III; Quinolone sensitive isolates 38 Ciprofloxacin 12 25 1

Levofloxacin 26 10 1 1

- Where is the statistical analysis that confirm the significant or non-significant correlation between the detected mutations and the resistance?

Author’s reply: Thank you for the question. Statistical analysis of significant parts of data has been added to the result section (Line 160 onwards). Correlation between detected mutations of both gyrA and parC, and level of tolerance to fluoroquinolones tested has been noted in text as p-values. Data as well as statistical analysis confirms the correlation as isolates with double mutation exhibited highest level of tolerance (MIC ≥128 µg/ml). I trust this is acceptable to you now.

Minor comments:

- Please check the language carefully in the whole manuscript e.g. Line101: "were" and Line 36.

Author’s reply: Thank you for the suggestion. The grammatical errors have been checked and corrected; I trust this is acceptable now.

- Clarify the abbreviation "QRDR” in line 56, as it the first time to be mentioned in introduction and please clarify other abbreviations.

Author’s reply: Thank you for the comment. All abbreviation, including "QRDR” in line 56 has been clarified now. I trust this is okay now.

- In line 61, does the word " motif" refer to the QRDR motif? please clarify.

Author’s reply: Thank you. Yes, the word “motif’ refers to the motif of QRDR region of GyrA, more specifically in P. aeruginosa, the 67-106th position of the GyrA subunit of DNA Gyrase serves as the QRDR region. Occurrence of mutation at this position leads to structural alteration of DNA Gyrase, changing the motif. This can result in reduced binding affinity of fluoroquinolones to DNA Gyrase, developing resistance. But to make it clearer, I have rephrased the sentence. I hope that clarifies the topic.

- In methods, please mention the ethical approval details.

Author’s reply: Thank you so much. The ethics statement was previously added at the end of the manuscript, and has been moved to the Methods section of the manuscript. I hope this is okay.

- Please carefully revise the whole manuscript as it contains some" review track changes" e,g line 78 ??

Author’s reply: Thank you. The manuscript has been revised as per your suggestion.

- Please mention the year of the CLSI version that was followed in this study in the references list.

Author’s reply: Thank you. In this study, the 31st edition of CLSI published in 2021 was followed. This has been mentioned in the references list [21] and in the methods section as well. I trust this is acceptable to you.

- In figure 1: Indicate the molecular size of different bands of the marker and also indicate its catalogue number

Author’s reply: Thank you for the suggestion. The molecular size of different bands as well as the amplified product is now indicated in the figure. Catalogue number of molecular markers used in this study is mentioned in the figure caption. I hope this is acceptable to you. The new figure with caption is:

Fig 1. PCR based detection of PMQR variants. Variants qnr gene sequences were PCR amplified using specific primers (Table 1). PCR products were electrophoresed through 1% agarose gels, stained with ethidium bromide, and visualized with UV light. The 100 bp DNA ladder (Promega, USA; Catalogue number G2101) indicates the size of PCR products. Lane 2, 6 and 10 represent variant-specific positive controls, and lane 3, 7 and 11 represent negative controls. No isolates were positive in qnrA, qnrB and qnrS specific PCR.

- In table 1:

• Write the name of genes correctly "Italic and small" in column 3 and in the whole manuscript.

• Put the "�C" only in the heading.

Author’s reply: Thank you for the suggestion. The table has been updated and I hope you’ll find it acceptable. The revised table is attached below:

Table 1. Primers used in this study

Gene Primer Sequence (5’ to 3’) Use Annealing Temperature (°C) Amplicon Size/ Nucleotide Position Reference

gyrA GAC GGC CTG AAG CCG GTG CAC Amplification and sequencing of gyrA subunit of Toposiomerase II 65 115-135 [22]

GCC CAC GGC GAT ACC GCT GGA 531-511

parC CGA GCA GGC CTA TCT GAA CTA T Amplification and sequencing of parC subunit of Toposiomerase IV 55 63-84

GAA GGA CTT GGG ATC GTC CGG A 366-344

qnrA AGA GGA TTT CTC ACG CCA GG Amplification of qnrA 54 580bp [23]

TGC CAG GCA CAG ATC TTG AC

qnrB CCT GAG CGG CAC TGA ATT TAT Amplification of qnrB 60 390bp [24]

GTT TGC TGC TCG CCA GTC GA

qnrS GCA AGT TCA TTG AAC AGG GT Amplification of qnrS 54 428bp [23]

TCT AAA CCG TCG AGT TCG GCG

fd1 TCT AAA CCG TCG AGT TCG GCG Amplification and sequencing of 16S rDNA 42 1500bp [25]

rp2 ACG GCT ACC TTG TTA CGA CTT

- - Write the name of protein correctly "Not Italic and start with capital letter" in Table 2 and in the whole manuscript when it refers to protein but not gene.

Author’s reply: Thank you for your comment. The names have been corrected accordingly and I trust this is acceptable.

---

## [Decision Letter · Decision Letter 1]

3 Dec 2024

PONE-D-24-13068R1Mutational alterations in the QRDR regions associated with fluoroquinolone resistance in Pseudomonas aeruginosa of clinical origin from Savar, DhakaPLOS ONE

Dear Dr. Rahman,

Thank you for submitting your manuscript to PLOS ONE. After careful consideration, we feel that it has merit but does not fully meet PLOS ONE’s publication criteria as it currently stands. Therefore, we invite you to submit a revised version of the manuscript that addresses the points raised during the review process. Please go through the Reviewers' comments very carefully and draft your responses accordingly.

We look forward to receiving your revised manuscript.

Kind regards,

Furqan Kabir

Academic Editor

PLOS ONE

Reviewers' comments:

Reviewer's Responses to Questions

**Comments to the Author**

1. If the authors have adequately addressed your comments raised in a previous round of review and you feel that this manuscript is now acceptable for publication, you may indicate that here to bypass the “Comments to the Author” section, enter your conflict of interest statement in the “Confidential to Editor” section, and submit your "Accept" recommendation.

Reviewer #3: (No Response)

Reviewer #4: All comments have been addressed

2. Is the manuscript technically sound, and do the data support the conclusions?

Reviewer #3: Partly

Reviewer #4: Yes

3. Has the statistical analysis been performed appropriately and rigorously? 

Reviewer #3: N/A

Reviewer #4: N/A

4. Have the authors made all data underlying the findings in their manuscript fully available?

Reviewer #3: Yes

Reviewer #4: Yes

5. Is the manuscript presented in an intelligible fashion and written in standard English?

Reviewer #3: No

Reviewer #4: Yes

6. Review Comments to the Author

Reviewer #3: This paper examines 53 clinical isolates of Pseudomonas aeruginosa (PA) for fluoroquinolone resistance and associated resistance mechanisms, including mutations in the GyrA and ParC genes as well as the presence of plasmid-mediated quinolone resistance (PMQR) genes. The research question and specific knowledge gaps are not clearly defined. Additionally, the detection of GyrA mutations in P. aeruginosa isolates from Bangladesh is not a new finding, hence, its contribution to the existing body of knowledge is limited. Moreover, the regional focus of the study limits the broader generalizability of its findings.

Although the sources of the isolates were broadly mentioned, the specific distribution of isolates across different clinical sample types was not provided, which could be important for identifying high-risk sample categories. Additionally, the procedures for the primary isolation of the organism were not described. Including population-related data, such as the number of patients with specific complications tested for the organism, would help estimate the prevalence of fluoroquinolone-resistant Pseudomonas aeruginosa. Furthermore, it is unclear why the study focuses exclusively on fluoroquinolone resistance, without addressing resistance to other antibiotics, which would provide a more comprehensive understanding of the resistance profile.

Specific comments:

• Please provide a brief description of the reconfirmation of isolates using standard culture methods.

• Although it was not mentioned but I assumed P. aeruginosa ATCC 27853 was used as a positive control. Did you use any negative control? If yes, please mention that too.

• How did you purify the PCR product. Please describe briefly.

• Why did you not test GyrB and ParE genes? Provide a reasonable justification.

• Which control strains did you use in tests for QRDR genes?

• The background section mentions other mechanisms of resistance; however, these were not investigated in the study. This creates gaps in understanding the complete resistance mechanisms among the isolates.

• Please replace the term “tolerance level” with “MIC value” throughout the manuscript.

• What was the rationale for testing only ciprofloxacin and levofloxacin? Why were other antibiotics within the same class not included in the analysis?

• Please provide the GenBank accession numbers for the sequences of all isolates.

• Which positive and negative controls were used in the sequencing analysis? Please also provide details of the control strains used in the PCR for detecting PMQR genes.

• Including a figure displaying the amino acid sequence alignment (multiple alignment) of the 15 isolates, particularly highlighting the regions with alterations, would improve the clarity of the findings.

• What are the current treatment guidelines for Pseudomonas aeruginosa infections? It would be valuable to frame the discussion around the clinical relevance of your study, particularly concerning fluoroquinolone resistance. If fluoroquinolones are not the drugs of choice, what are the implications of your findings?

Reviewer #4: 1- the authors used statics in the paper but didn't mention the paragraph belong to it.

2- the figure 2 repeated two times.

3- in figure two the author claim in some lines no results but each lines has some band? this confused results specially found line with negative results.

4- the paper didn't has study limitation

7. PLOS authors have the option to publish the peer review history of their article (what does this mean? ). If published, this will include your full peer review and any attached files.

**Do you want your identity to be public for this peer review?** For information about this choice, including consent withdrawal, please see our Privacy Policy .

Reviewer #3: No

Reviewer #4: **Yes: ** Sawsan Mohammed kareem

---

## [Author Response · Author response to Decision Letter 2]

8 Jan 2025

Review Comments to the Author:

Reviewer #3:

• The research question and specific knowledge gaps are not clearly defined.

Author’s Reply: Thank you for the comment. Detailed information focusing on research question is now included in the Introduction section of the manuscript. We believe this is acceptable now.

• Additionally, the detection of GyrA mutations in P. aeruginosa isolates from Bangladesh is not a new finding, hence, its contribution to the existing body of knowledge is limited. Moreover, the regional focus of the study limits the broader generalizability of its findings.

Author’s Reply: Thank you for the question. While we agree that GyrA mutations in P. aeruginosa is not a new finding, we also believe that surveillance of antimicrobial resistance and resistance determinants are crucial to understand trends in AMR. The present study also focuses on the dissemination of PMQR genes in high-complexity hospitals, as they can serve as a reservoir for mobile genetic elements. We believe regional data is an integral part of global AMR surveillance, and contributions like our study helps to understand recent trends in AMR globally.

• Although the sources of the isolates were broadly mentioned, the specific distribution of isolates across different clinical sample types was not provided, which could be important for identifying high-risk sample categories. Additionally, the procedures for the primary isolation of the organism were not described. Including population-related data, such as the number of patients with specific complications tested for the organism, would help estimate the prevalence of fluoroquinolone-resistant Pseudomonas aeruginosa.

Author’s Reply: Thank you for the comment. Data regarding distribution of isolates and population related data have been previously published in an article that described the source of the isolates used in this study (Anjum et al, 2023). Hence the isolates are labeled as “Laboratory Collection” in the present manuscript. We have refrained from including the same data in this manuscript, as well as the procedures of primary isolation and population related data, to avoid repetition and redundancy. However, based on your suggestion, we will be attaching the sample distribution and population related data as supplementary data. We believe this is okay and hope is acceptable to you.

• Furthermore, it is unclear why the study focuses exclusively on fluoroquinolone resistance, without addressing resistance to other antibiotics, which would provide a more comprehensive understanding of the resistance profile.

Author’s Reply: Thank you for your question. The present investigation is part of an ongoing project that aims to investigate the multidrug resistance in clinical infection caused by different pathogenic bacteria, including P. aeruginosa from Savar, Bangladesh. It also studies the resistance mechanisms to different groups of clinically important antibiotics including β-lactams, carbapenems and fluoroquinolones. Since the data obtained is large in quantity and would be easy to describe separately, we chose to prepare manuscripts focusing different antibiotics. Our previous studies have already discussed the resistance determinants to β-lactams (Anjum et al, 2023) and carbapenems among the P. aeruginosa isolates used in this study. The mentioned article addresses the MDR nature of the isolates and discusses their AMR profiles. The current manuscript, therefore, is solely focused to showcase data regarding resistance to fluoroquinolones in P. aeruginosa from Savar, Dhaka. We hope this is acceptable to you.

Specific comments:

• Please provide a brief description of the reconfirmation of isolates using standard culture methods.

Author’s Reply: Thanks for the suggestion. A brief description of standard culture method for identification of P. aeruginosa is now included in the Materials and Methods section of the manuscript. We believe this is okay now.

• Although it was not mentioned but I assumed P. aeruginosa ATCC 27853 was used as a positive control. Did you use any negative control? If yes, please mention that too.

Author’s Reply: Thank you for your question. Yes, P. aeruginosa ATCC 27853 was used as positive control. And as negative control, E. coli ATCC 25922 was used in this study. This information is now mentioned in the text. We trust this is okay now.

• How did you purify the PCR product. Please describe briefly.

Author’s Reply: Thank you for the question. PCR products were purified using Wizard® SV Gel and PCR Clean-Up System from Promega, USA (Catalog Number: A9281). The clean-up system is now mentioned in the text. We believe this is acceptable now.

• Why did you not test GyrB and ParE genes? Provide a reasonable justification.

Author’s Reply: Thank you for your question. Although our initial plan was to investigate mutational alterations in the relevant subunits of both Topoisomerase II (GyrA and GyrB) and Topoisomerase IV (ParC and ParE), the study was limited financially. Since we had limited funding, we chose to focus on the major subunits of both enzymes (GyrA and ParC). These subunits are more prone to alterations and are more frequently associated with fluoroquinolone resistance, so we chose to investigate these first. We are hoping to expand the present study by investigating GyrB and ParE alongside other mutational alterations associated with multidrug resistance like nfxC, nfxB and mexR once further fundings are available. We hope this explains your query.

• Which control strains did you use in tests for QRDR genes?

Author’s Reply: Thank you for the question. For PCR and sequence analysis of QRDR genes, P. aeruginosa PAO1 was used as a control strain. Obtained sequences were compared with associated P. aeruginosa PAO1 loci from NCBI GenBank as reference. We hope this answers your question.

• The background section mentions other mechanisms of resistance; however, these were not investigated in the study. This creates gaps in understanding the complete resistance mechanisms among the isolates.

Author’s Reply: Thank you for the comment. In addition to mutational alterations in QRDR regions and acquisition of PMQR genes, fluoroquinolone resistance can be mediated by overexpression of efflux pumps like MexAB-OprM. We have also explored the phenotypic overexpression of efflux pump in efflux inhibitor induced MIC depression assay, but didn’t find any significant efflux pump activity. This has been discussed in the discussion section (Line 214 onwards). We hope this is acceptable to you.

• Please replace the term “tolerance level” with “MIC value” throughout the manuscript.

Author’s Reply: Thank you for the suggestion. We have replaced “tolerance level” with “MIC value” throughout the manuscript in relevant contexts. We trust this is okay now.

• What was the rationale for testing only ciprofloxacin and levofloxacin? Why were other antibiotics within the same class not included in the analysis?

Author’s Reply: Thank you for the question. Fluoroquinolones like ciprofloxacin and levofloxacin are among the most commonly prescribed and therapeutically used antibiotics of the class. Ciprofloxacin is a second-generation fluoroquinolone, while levofloxacin belongs from the third generation of the class. These antibiotics are the most frequently prescribed fluoroquinolones at the hospitals the isolates are obtained from, and that is the main rationale for selecting these antibiotics for the current study. We hope this answers your question.

• Please provide the GenBank accession numbers for the sequences of all isolates.

Author’s Reply: Thank you for the suggestion. The present study investigated the alterations of gyrA and parC subunits for mutational alterations. Since all of the isolates had similar alterations in their gyrA and no novel mutations were identified from the sequence obtained, we decided to submit the sequence from a representative isolate. Similarly, the parC sequence obtained from a representative isolate is submitted to the GenBank. Their accession numbers PP501828 (gyrA) and PP526740 (parC) are already mentioned in this study. Sequences of all isolates are not submitted to the GenBank to avoid repetition/ redundancy, but are now available as supplementary data. We hope this is acceptable to you.

• Which positive and negative controls were used in the sequencing analysis? Please also provide details of the control strains used in the PCR for detecting PMQR genes.

Author’s Reply: Thank you for the question. For sequence analysis, P. aeruginosa PAO1 was used as a positive control and obtained sequences were compared with associated P. aeruginosa PAO1 loci from NCBI GenBank as reference. In PCR for PMQR gene detection, locally curated control strains E. coli DMC1 (for qnrA and qnrB) and E. coli Shishu3 (for qnrS) was used as positive control. These control strains were gift from icddr.b (International Centre for Diarrheal Disease Research, Bangladesh), and have been confirmed to harbor the mentioned PMQR genes. The controls are now mentioned in the manuscript. We believe this is okay now.

• Including a figure displaying the amino acid sequence alignment (multiple alignment) of the 15 isolates, particularly highlighting the regions with alterations, would improve the clarity of the findings.

Author’s Reply: Thank you for the proposition. We believe we have already included two figures displaying the alignment of amino acid sequences highlighting regions with alterations in the manuscript (Fig 1: Sequence analysis of gyrA and parC showing alterations). We trust that is okay.

• What are the current treatment guidelines for Pseudomonas aeruginosa infections? It would be valuable to frame the discussion around the clinical relevance of your study, particularly concerning fluoroquinolone resistance. If fluoroquinolones are not the drugs of choice, what are the implications of your findings?

Author’s Reply: Thank you for the suggestion. We have included a section in the Discussion part of the manuscript explaining the importance of fluoroquinolones from a therapeutic point-of-view, and why it is relevant to study fluoroquinolone resistance in P. aeruginosa. We trust this is acceptable now.

Reviewer #4:

1- the authors used statics in the paper but didn't mention the paragraph belong to it.

Author’s Reply: Thank you for the suggestion. A subsection titled “Statistical Analysis” is now included in the Materials and Methods section of the manuscript. We trust this is okay now.

2- the figure 2 repeated two times.

Author’s Reply: Thank you for the enquiry. However, we could not find the repetition mentioned in the manuscript. We believe we have submitted the figure separately and it’s not included in the manuscript file.

3- in figure two the author claim in some lines no results but each lines has some band? this confused results specially found line with negative results.

Author’s Reply: Thank you for the question. The authors claim the results in figure two as negative since no specific amplification is observed in the mentioned lanes. Amplification often produces non-specific amplicons that can be observed as faint bands in places not corresponding to the actual amplicons. Hence, positive controls are used and non-specific amplifications are considered as negative result. We hope this clears the confusion and is acceptable now.

4- the paper didn't has study limitation

Author’s Reply: Thank you for the comment. Limitations of the present study is now discussed briefly in the Discussion section of the manuscript. We hope this is acceptable to you.

---

## [Editor Report · Decision Letter 2]

29 Jan 2025

Mutational alterations in the QRDR regions associated with fluoroquinolone resistance in Pseudomonas aeruginosa of clinical origin from Savar, Dhaka

PONE-D-24-13068R2

Dear Dr. Rahman,

We’re pleased to inform you that your manuscript has been judged scientifically suitable for publication and will be formally accepted for publication once it meets all outstanding technical requirements.

Kind regards,

Furqan Kabir

Academic Editor

PLOS ONE
---

## [Editor Report · Acceptance letter]

PONE-D-24-13068R2

PLOS ONE

Dear Dr. Rahman,

I'm pleased to inform you that your manuscript has been deemed suitable for publication in PLOS ONE. Congratulations! Your manuscript is now being handed over to our production team.

Kind regards,

on behalf of

Dr. Furqan Kabir

Academic Editor

PLOS ONE